# Cohort profile: The 'Children's Health in Care in Scotland' (CHiCS) study—a longitudinal dataset to compare health outcomes for care experienced children and general population children

Mirjam Allik  ,[1] Denise Brown,[1] Courtney Taylor Browne Lūka,[2] Cecilia Macintyre,[3] Alastair H Leyland,[1] Marion Henderson[4]

[1]MRC/CSO Social and Public Health Sciences Unit, University of Glasgow, Glasgow, UK
[2]School of Psychology, University of Glasgow, Glasgow, UK
[3]Data for Research Unit, Scottish Government, Edinburgh, UK
[4]School of Social Work & Social Policy, University of Strathclyde, Glasgow, UK

**Correspondence to**
Dr Mirjam Allik;
mirjam.allik@glasgow.ac.uk

## ABSTRACT

**Purpose** The Children's Health in Care in Scotland Cohorts were set up to provide first population-wide evidence on the health outcomes of care experienced children (CEC) compared with children in the general population (CGP). To date, there are no data on how objective health outcomes, mortality and pregnancies for CEC are different from CGP in Scotland.

**Participants** The CEC cohort includes school-aged children who were on the 2009/2010 Scottish Government's Children Looked After Statistics (CLAS) return and on the 2009 Pupil Census (PC). The children in the general population cohort includes those who were on the 2009 PC and not on any of the CLAS returns between 1 April 2007 and 31 July 2016.

**Findings to date** Data on a variety of health outcomes, including mortality, prescriptions, hospitalisations, pregnancies, and Accident & Emergency attendances, were obtained for the period 1 August 2009 to 31 July 2016 for both cohorts. Data on socioeconomic status (SES) for both cohorts were available from the Birth Registrations and a small area deprivation measure was available from the PC. CEC have, on average, lower SES at birth and live in areas of higher deprivation compared with CGP. A higher proportion of CEC have recorded events across all health data sets, and they experienced higher average rates of mortality, prescriptions and hospitalisations during the study period. The reasons for contacting health services vary between cohorts.

**Future plans** Age-standardised rates for the two cohorts by sex and area deprivation will be calculated to provide evidence on population-wide prevalence of main causes of death, reasons for hospitalisation and types of prescription. Event history analysis will be used on matched cohorts to investigate the impact of placement histories and socioeconomic factors on health.

## INTRODUCTION

There is very little quantitative evidence on how the health of care experienced children[i]

---

[i]'Looked after children' is the statutory term for children in the care of local authorities, but often this excludes children who have left care. The term 'care experienced children' is now widely used to describe any person who

---

**Strengths and limitations of this study**

► The CHiCS (Children's Health in Care in Scotland) project is the first population-wide longitudinal data collection in the UK that links administrative data on social care, births, deaths, hospitalisations and prescriptions to compare the health of care experienced children (CEC) with children in the general population (CGP).

► The study looks at a wide range of health outcomes, including inpatient and outpatient hospitalisations, prescriptions, accidents & emergency attendances, pregnancies and mortality.

► CEC had higher average rates of mortality, prescriptions and hospitalisations during the study period compared with CGP, and the differences in health between the two cohorts are most notable for mental, sexual and reproductive health.

► The weaknesses of the study include errors in the individual data sets and in data linkage (affecting about 3% of the study population), and the exclusion of children who were not in school or were in independent schools (approximately 4% of all school-aged children).

---

(CEC) compares with other children in Scotland and in the UK. Invariably, the evidence that is available suggests that care experience is related to poorer health,[1–4] but often this is based on small sample sizes or without comparison with children who have not been in care.[2 3] The only population-wide evidence on the effects of care experience on health in Scotland (and the UK) comes from a linked administrative data study on dental health.[4] The study indicates considerable complexity in how different care experiences are related

---

has experience of being in care, regardless of their placement length, type or age.

**Table 1** Number, percentage and length of placements during study period by the type of placement

| Placement type | Placements | | Length (months) | |
|---|---|---|---|---|
| | N | % | Mean | Median |
| Private household placement | | | | |
| At home | 8716 | 31.2 | 16.7 | 12 |
| With foster carers provided by LA | 5705 | 20.4 | 23.6 | 13 |
| With foster carers purchased by LA | 2362 | 8.4 | 25.4 | 15 |
| With friends/relatives | 4958 | 17.7 | 25.6 | 17 |
| In other community | 349 | 1.2 | 6.4 | 4 |
| With prospective adopters | 180 | 0.6 | 15.2 | 12 |
| Residential placements | | | | |
| In LA home | 2545 | 9.1 | 11.2 | 6 |
| In residential school | 1051 | 3.8 | 16.3 | 12 |
| In secure accommodation | 729 | 2.6 | 4.8 | 3 |
| In voluntary home | 352 | 1.3 | 13.9 | 8 |
| In crisis care | 85 | 0.3 | 6.0 | 2 |
| Other residential | 929 | 3.3 | 10.9 | 5 |
| Total | 27 961 | 100 | 19.2 | 11 |

LA, local authority.

to health. Overall, CEC experience worse dental health outcomes compared with children who have not been in care, but there are significant differences in health outcomes by care type. For example, urgent and non-urgent dental needs were highest for those in home or kinship care and lower for children in foster care.

The cited research also shows that CEC are more likely to live in deprived areas compared with those who have not been in care, with half of all children in home and kinship care living in the most deprived areas. Importantly, differences in health outcomes between children persist after accounting for area deprivation.[4] Many previous studies have often not been able to account for area deprivation and family socioeconomic status (SES),[1] something that has a substantial effect on both health outcomes[5] and the chances of experiencing social care.[6]

In the UK, researchers have reported higher mortality,[7] poorer self-rated general and mental health,[8] and higher rates of some physical illnesses (epilepsy, cystic fibrosis and cerebral palsy)[9] among CEC. There is also evidence of higher pregnancy rates among young women in care.[10] Research in Sweden, Finland and Canada underline high rates of mental health–related problems like suicide and suicide attempts, psychiatric disorders and substance abuse among the care experienced population.[11–15] Other international research has shown evidence of higher rates of avoidable deaths (eg, from causes that could have been prevented by timely medical care, or homicide and unintentional injuries)[16] and emergency department visits[17] among children in foster care.

Beyond this, there is very limited evidence on which objective health outcomes are most likely to differ between children who have and who have not experienced care

and for this reason our study includes a wide variety of health measures, including hospitalisations, clinic attendances, prescriptions and mortality. Based on these findings and our discussions with the Centre for Excellence for Children's Care and Protection (CELCIS) and the Scottish Government, we also decided to include data on pregnancies for the young women in our cohort.

Administrative data linkage is the only feasible way of comparing a wide range of objective health outcomes between CEC and children in the general population (CGP) nationally, and in a representative way. Very few available data sets in Scotland and in the UK include details of both out of home or formal care and indicators of health. National surveys of children's health (eg, Health Behaviour in School-aged Children study) and Scotland's Census include indicators of out-of-home and residential care, but these sources do not include any information on care history, or placement types and lengths. In Scotland, a child can also be 'looked after' while living at home with their parent(s) under a home supervision order. For survey data, this means that it is not always possible to distinguish whether the child is formally considered 'looked after' while living at home or with relatives. Finally, some surveys have too few CEC to analyse health outcomes for this subpopulation.

The lack of quantified evidence on the health of CEC has long been recognised as a major obstacle to evidence-based policy-making in the field.[18] The Children's Health in Care in Scotland (CHiCS) study provides the first robust nationwide evidence, with longitudinal data on care histories for CEC and a wide range of health outcomes for both cohorts of children. Improved knowledge about the health outcomes of CEC, particularly in comparison

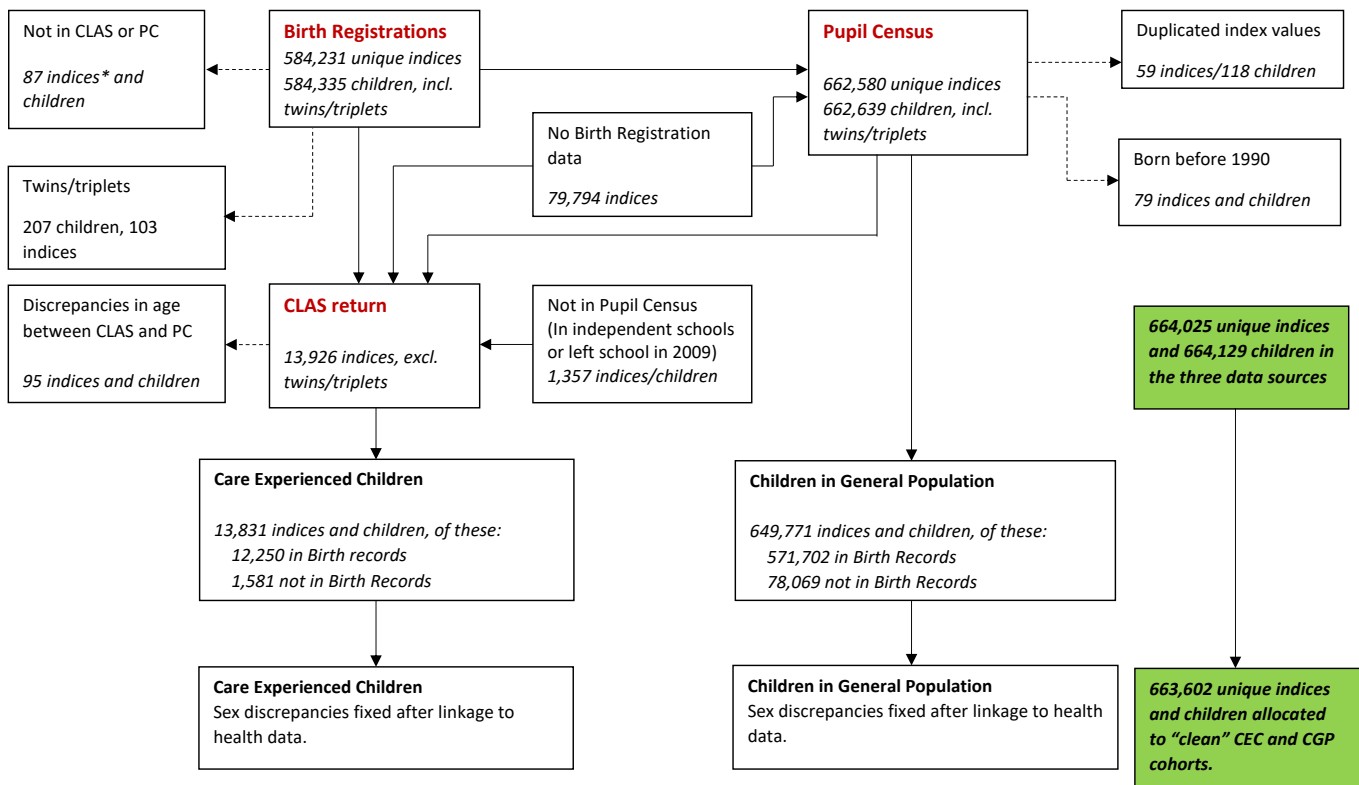

**Figure 1** Development and size of the two study cohorts. *Each index refers to a unique child. When there are more index values than children in the data (in Birth Registrations and Pupil Census) this means that more than one child had the same linkage identifiers (date of birth, postcode, sex). This is the case for twins/triplets and other linkage errors, these duplicated index values were removed from the data as it was not possible to determine which health records belonged to which child. Note: Dashed arrows indicate children who were removed from the cohorts during data cleaning.

with CGP, will assist with the allocation of services by identifying precise areas of heightened needs, and will inform future research seeking to understand health inequalities between CEC and CGP.

### Cohort description

The CHiCS cohorts follow a range of health outcomes for two groups of school-aged children in publicly funded schools in Scotland: CEC and CGP (ie, not care experienced). Children in Scotland usually start primary school at age 4.5 to 5.5 years. Secondary school begins at age 11 or 12 for a compulsory 4 years with the following 2 years being optional. All children in the 2009 Pupil Census (PC), taken on 21 September, were included in the baseline cohorts. The CEC cohort was then defined as all those children who were also on the 2009/2010 Scottish Government's Children Looked After (CLAS) return, collected between 1 August 2009 and 31 July 2010. The care histories of CEC were followed up to 31 July 2016 longitudinally, linking seven CLAS returns. The CGP cohort was defined as children who were on the 2009 PC that have not experienced care, that is, not appeared on any of the CLAS returns from the first collection date on 1 April 2007 to the most recent date included in our study on 31 July 2016. The latter comparison was made to all possible CLAS returns available at the time to ensure that

none of the general population children were or became looked after during the study.

The two cohorts of children were linked to all other data sources using the Scottish Candidate Number (SCN) present on the PC. The linkage of CLAS records to health is only possible via the SCN, and CEC who were not in school in 2009/2010, or had an invalid or missing SCN could not be included in the cohort. The 2009/2010 CLAS return was chosen as the baseline cohort as the completeness of SCN on the CLAS return became high enough to make linkage between the CLAS return, the PC and health data representative. Details of SCN completeness for a single CLAS return has been published previously[19] and for our study this was improved further by the longitudinal linkage of multiple CLAS returns. The PC itself has nearly 100% coverage of all children in publicly funded schools, is frequently used in research and in provision of Accredited National Statistics.

Health outcomes for the CHiCS cohorts will be examined over the 7-year period between 2009 and 2016. The health data include the number of and diagnoses relating to hospital admissions (from Scottish Morbidity Records; SMR01/02/04) or specialty for outpatient clinic attendance (SMR00), number of and British National Formulary (BNF) chapter and section code for prescriptions

**Table 2** Individual-level socioeconomic profiles of CEC and CGP cohorts

| | Children with birth records | | | | All children | | | |
|---|---|---|---|---|---|---|---|---|
| | CGP | | CEC | | CGP | | CEC | |
| | N | % | N | % | N | % | N | % |
| No (%) of children | 571 702 | 88.0 | 12 250 | 88.6 | 649 771 | 100.0 | 13 831 | 100.0 |
| Female | 281 221 | 49.2 | 5565 | 45.4 | 319 438 | 49.2 | 6274 | 45.4 |
| Male | 290 481 | 50.8 | 6685 | 54.6 | 330 333 | 50.8 | 7557 | 54.6 |
| Age in months | | | | | | | | |
| Mean (SD) | 130 (43.0) | | 135 (42.2) | | 131 (43.0) | | 136 (42.3) | |
| Year of birth | | | | | | | | |
| 1990–1995 | 160 644 | 28.1 | 4114 | 33.6 | 184 923 | 28.5 | 4830 | 34.9 |
| 1996–2000 | 234 120 | 41.0 | 4862 | 39.7 | 265 453 | 40.9 | 5413 | 39.1 |
| 2001–2004 | 176 938 | 30.9 | 3274 | 26.7 | 199 395 | 30.7 | 3588 | 25.9 |
| For children with birth records: | | | | | | | | |
| Mother's age at birth | | | | | | | | |
| Mean (SD) | 28.3 (5.7) | | 24.5 (5.8) | | | | | |
| Aged under 20 | 43 624 | 7.6 | 2792 | 22.8 | | | | |
| Aged 20–24 | 105 653 | 18.5 | 4066 | 33.2 | | | | |
| Aged 25–29 | 175 114 | 30.6 | 2898 | 23.7 | | | | |
| Aged 30–34 | 167 634 | 29.3 | 1664 | 13.6 | | | | |
| Aged 35–39 | 68 442 | 12.0 | 682 | 5.6 | | | | |
| Aged 40+ | 10 737 | 1.9 | 130 | 1.1 | | | | |
| Missing | 498 | 0.1 | 18 | 0.1 | | | | |
| Parental employment status at birth | | | | | | | | |
| Parental (for births 1990–1995) | | | | | | | | |
| Employee | 117 526 | 73.2 | 2715 | 66.0 | | | | |
| Manager | 14 836 | 9.2 | 84 | 2.0 | | | | |
| Supervisor | 4788 | 3.0 | 35 | 0.9 | | | | |
| Self-employed (with employees) | 4919 | 3.1 | 27 | 0.7 | | | | |
| Self-employed (without employees) | 6955 | 4.3 | 67 | 1.6 | | | | |
| Student/unemployed/not available | 11 620 | 7.2 | 1186 | 28.8 | | | | |
| Missing | 0 | 0.0 | 0 | 0.0 | | | | |
| Mother (for births 1996–2004) | | | | | | | | |
| Employee | 271 711 | 66.1 | 3138 | 38.6 | | | | |
| Manager | 28 201 | 6.9 | 58 | 0.7 | | | | |
| Supervisor | 9315 | 2.3 | 39 | 0.5 | | | | |
| Self-employed (with employees) | 3811 | 0.9 | 7 | 0.1 | | | | |
| Self-employed (without employees) | 7558 | 1.8 | 33 | 0.4 | | | | |
| Student/unemployed/not available | 90 425 | 22.0 | 4861 | 59.7 | | | | |
| Missing | 37 | 0.0 | 0 | 0.0 | | | | |
| Father (for births 1996–2004) | | | | | | | | |
| Employee | 274 748 | 66.8 | 4467 | 54.9 | | | | |
| Manager | 44 524 | 10.8 | 130 | 1.6 | | | | |
| Supervisor | 12 791 | 3.1 | 61 | 0.7 | | | | |
| Self-employed (with employees) | 15 036 | 3.7 | 79 | 1.0 | | | | |

Continued

**Table 2** Continued

| | Children with birth records | | | | All children | | | |
|---|---|---|---|---|---|---|---|---|
| | CGP | | CEC | | CGP | | CEC | |
| | N | % | N | % | N | % | N | % |
| Self-employed (without employees) | 24 631 | 6.0 | 285 | 3.5 | | | | |
| Student/unemployed/not available | 15 889 | 3.9 | 1227 | 15.1 | | | | |
| Missing | 23 439 | 5.7 | 1887 | 23.2 | | | | |

CEC, care experienced children; CGP, children in the general population.

(PIS), and cause of death (online supplemental table 1). For Accidents & Emergency (A&E) data, only the number of attendances per year for each child was available. All the SMR, PIS and A&E data are collected and shared by Public Health Scotland. The Birth and Death Registrations data are available from the National Records of Scotland. All included health, birth and death data have been widely used in research, undergo regular data quality checks and are considered high quality.[20–22]

For hospitalisations, we also have access to outcomes from 1990 to account for past health. Previous research has identified differences in health service use (in primary care) among both mothers and CEC before the child entered care.[23] For A&E data, past events are recorded from 2007 but the individual-level prescribing data are only available from 2009.[20] For SMR02 and SMR04, admission reason was also recorded and will be used in research if the quality permits.

For CEC, we have data on all care placements during the study period, including the start and end month and year of the placements (giving the length in months) and the care placement type. The majority of children (58%) had one care placement during the study, 18.5% had two, 9.5% three and 5.4% four placements, leaving 8.4% with five or more placements during the 7-year period. Table 1 shows that the most common placement types were at home under a Supervision Requirement (regular contact with social services) (31.2%), in foster care (total 28.8%) or with friends/relatives (17.7%). The mean placement length is 19 months (just over 1.5 years), but this varies considerably across placement types, with residential types generally having shorter placement lengths compared with living within a private (family) household (table 1). The data also include an indicator if the child was in care before the study start date and the length and type of these placements. However, the quality of the earlier records is inconsistent and will require more analysis before we can determine if and how this can be used in research.

The individual-level linkage process of the 10 data sources is described in online supplemental figure 1 and follows the steps outlined previously for research on dental health.[19] Data cleaning steps and the size of the two cohorts are described in figure 1. The key data sets in deriving the cohorts were the CLAS return and the PC.

Twins and triplets were removed using information from Birth Registrations (the linkage is based on postcode, date of birth and sex, thus twins living at the same address are difficult to reliably link).

In total, 663 602 school-aged children were included in the study. Children were aged 4 to 19 years at the start of the study and 11 to 26 years by the end of follow-up in 2016. In total, 13 831 (2.1%) were identified as CEC and 649 771 (97.9%) as CGP. There was a higher proportion of male CEC than male CGP (54.6% compared with 50.8%; see table 2). On average, CEC tended to be older than CGP. CEC were aged 11 years and 3 months on average (mean age in months=135, SD 42.2) and CGP aged 10 years and 10 months on average (mean age in months=130, SD 43.0), with a higher proportion of children in the CEC cohort born in the early 1990s. Differences in the age and sex distributions for all CEC and GCP with birth records are comparable with the differences in the age and sex distributions for all CEC and CGP. This suggests that children with birth records available are representative of all children in the study.

### Individual-level socioeconomic profile
In total, 88.6% and 88.0% of CEC and GCP, respectively, had birth records available with information including mothers' age at the time of birth and parental employment status (see table 2, columns on data for children with birth records). On average, mothers of CEC were younger at birth (24.5 years old compared with 28.3 years). In total, 56.1% of mothers of CEC were aged under 25 years, compared with 26.1% of mothers of CGP.

For children born before 1996, only one parent's occupation was recorded at birth (father's occupation if married, otherwise mother's occupation) and we therefore report parental employment status for children born before 1996. Parents of CEC were less likely to be in employment than parents of CGP (71.2% in employment compared with 92.8%). From 1996 onward, both mother's and father's occupation were recorded for all births registered by married couples or for births that were jointly registered by unmarried couples, so we report both mother's and father's employment status. Only 40% of mothers of CEC, born after 1995, were in employment compared with 78% of mothers of CGP. Fathers of CEC were also less likely to be in employment than fathers of CGP. Note

**Table 3** Deprivation level of area of residence of CEC and CGP cohorts

| | All children with birth records | | | | | | | | All children | | | |
| | Area of residence at birth | | | | Area of residence in 2009 | | | | Area of residence in 2009 | | | |
| | CGP | | CEC | | CGP | | CEC | | CGP | | CEC | |
| | N | % | N | % | N | % | N | % | N | % | N | % |
|---|---|---|---|---|---|---|---|---|---|---|---|---|
| **No (%) of children** | 571702 | **88.0** | 12250 | **88.6** | 571702 | **88.0** | 12250 | **88.6** | 649771 | **100.0** | 13831 | **100.0** |
| Quintile of deprivation | | | | | | | | | | | | |
| Q1 (most deprived) | 142253 | 24.9 | 7166 | 58.5 | 123331 | 21.6 | 5061 | 41.3 | 140120 | 21.6 | 5591 | 40.4 |
| Q2 | 116146 | 20.3 | 2650 | 21.6 | 112764 | 19.7 | 2636 | 21.5 | 125764 | 19.4 | 2914 | 21.1 |
| Q3 | 107180 | 18.7 | 1420 | 11.6 | 110157 | 19.3 | 1641 | 13.4 | 125704 | 19.3 | 1852 | 13.4 |
| Q4 | 103610 | 18.1 | 694 | 5.7 | 112379 | 19.7 | 1116 | 9.1 | 129566 | 19.9 | 1268 | 9.2 |
| Q5 (least deprived) | 101665 | 17.8 | 303 | 2.5 | 112846 | 19.7 | 613 | 5.0 | 127979 | 19.7 | 696 | 5.0 |
| *Missing* | 858 | 0.2 | 17 | 0.1 | 225 | 0.0 | 1183 | 9.7 | 638 | 0.1 | 1510 | 10.9 |

CEC, care experienced children; CGP, children in the general population.

also that the occupational status of CEC fathers was more likely to be missing (23.2% compared with 5.7%) indicating more absence among CEC fathers.

### Area-level socioeconomic profile

Birth records include information on area of residence at birth (2001 data zone) and the PC records information on area of residence (2001 data zone) at the start of the study in 2009. We linked data zones at birth to the 2004 Scottish Index of Multiple Deprivation (SIMD) and linked data zones at the start of the study in 2009 to SIMD 2009v2. Table 3 gives the SIMD quintiles for area of birth for all children with birth records, for area of residence at start of study in 2009 for all children with birth records, and for area of residence at start of study in 2009 for all children in the study. In total, 59% of CEC (with birth records) lived in the most deprived quintile of deprivation compared with 25% of CGP at birth. By 2009, 41% of CEC (with birth records) were living in the most deprived quintile, perhaps reflecting residential moves into a less deprived area following birth and being taken into care.

The data zones can also be linked to Health Boards and to the Scottish Government Urban Rural Classification 2003/2004 (at birth) and 2009/2010 (in 2009), used to classify small areas as urban, rural or remote (online supplemental table 2). Scotland is divided into 14 Health Boards which have responsibility for health protection, promotion and the delivery of services to their population. Social care is the responsibility of local government through Scotland's 32 local authorities. In terms of the urban–rural classification, there was a shift for CEC away from large urban areas between birth and 2009 (50% in large urban areas at birth compared with 40% in 2009 for CEC with birth records). Perhaps unsurprisingly, CEC were also more likely to have changed data zones between birth and the start of the study in 2009 (online supplemental table 3). Only 12.4% of CEC were living in the same data zone of residence at birth by the beginning of the study. CEC were also more likely to have missing information on data zone of residence, highlighting one of the difficulties in studying hard-to-reach populations. In total, 11% of CEC did not have residential details recorded in the Pupil Census, compared with just 0.04% in the general population.

### National comparisons

Our cohort of CEC include those who were on the 2009/2010 CLAS return and in the 2009 PC. To give an estimate of how well we capture the whole population of CEC and CGP in Scotland, we compare all children in our CEC cohort with Scottish Government National Children's Social Work Statistics (online supplemental table 4). Compared with national statistics on children in care, we see a very similar sex distribution with around 55% males and 45% females in care. Our age distribution varies in the youngest age group (0–4 years) as we include only school-aged children in our cohort and therefore only have a small proportion of 4-year-olds who were

**Table 4** Main health outcomes for the two cohorts, 2009–2016

| | Children with at least one event | | | | Mean per child | | Rate* | | Ratio of rates |
|---|---|---|---|---|---|---|---|---|---|
| | CGP | | CEC | | CGP | CEC | CGP | CEC | CEC:CGP |
| | N | %† | N | % | | | | | |
| Total children | 649771 | 100 | 13831 | 100 | | | | | |
| Total female | 319438 | 49.2 | 6274 | 45.4 | | | | | |
| Deaths | 746 | 0.1 | 78 | 0.6 | 0.11 | 0.56 | 15.3 | 83.5 | 5.48 |
| PIS | 603628 | 92.9 | 12597 | 91.1 | 28.32 | 34.71 | 3642.3 | 4446.0 | 1.22 |
| SMR00 | 382590 | 58.9 | 9427 | 68.2 | 6.30 | 8.09 | 469.7 | 736.1 | 1.57 |
| SMR01 | 179551 | 27.6 | 5404 | 39.1 | 2.22 | 2.60 | 94.0 | 150.5 | 1.60 |
| SMR02‡ | 14269 | 4.5 | 1302 | 20.8 | 2.82 | 3.24 | 36.6 | 158.5 | 4.33 |
| SMR04 | 2197 | 0.3 | 323 | 2.3 | 3.56 | 2.90 | 179.2 | 923.2 | 5.15 |
| A&E | 434528 | 66.9 | 10826 | 78.3 | 3.36 | 5.86 | 273.2 | 572.1 | 2.09 |

SMR00—Outpatient Attendance; SMR01—General/Acute Inpatient and Day Case; SMR02—Maternity Inpatient and Day Case; SMR04—Mental Health Inpatient and Day Case.

*Age standardised for ages 0–24 using the 2013 European standard population per 1000 person-years (PY). For deaths and SMR04, rates are shown per 100 000 PY.

†Percentage from cohort.

‡Calculated for female members of the cohort.

A&E, Accident & Emergency; CEC, care experienced children; CGP, children in the general population.

registered for school. For ages 5–15, we are capturing very similar numbers of CEC compared with the published national statistics.

We also compared all children in our study (CEC and CGP; n=663602) to Scotland's 2011 census population aged 0–19 years (online supplemental table 5). The comparison with the 2011 population census shows we have captured a high proportion of children of compulsory school age (95.5% of children 5–11 years old and 93.4% of children 12–15 years old). Note that for ages 5–15, we are including about 95% of the population in Scotland. We also have captured 98.1% of the 2009 Pupil Census population of 676740 pupils.[24] In conclusion, these comparisons suggest that our two cohorts are capturing a very high proportion of all school-aged children in Scotland and our study is representative of this population.

### Patient and public involvement statement

We collaborated with CELCIS when planning and designing this research project. We have set up an Advisory Group including representatives from children's charities and public authorities responsible for the welfare of children and CEC to help guide and contextualise the research, and undertake knowledge exchange and user engagement programme. The planned knowledge exchange and user engagement programme will act as patient and public involvement and will be undertaken as the research progresses in 2021/2022.

### Findings to date

The descriptive analysis of the data in table 4 shows that the proportion of CEC who have had at least one event recorded in any of the health data sources is higher compared with CGP for all data sets except for prescriptions. The biggest differences are evident for deaths, with CEC have 5.5 times higher mortality compared with CGP (CEC$_{rate}$=83.5, CGP$_{rate}$=15.3), and hospitalisations for maternity (CEC have 4.3 times higher rates in SMR02; CEC$_{rate}$=158.5, CGP$_{rate}$=36.6) and mental health inpatient and day cases (CEC have 5.2 times higher rates in SMR04; CEC$_{rate}$=923.2, CGP$_{rate}$=179.2). CEC also experience substantially more health events (hospital visits, prescriptions) per child in all health data sets, with the biggest differences evident for SMR02, SMR04 and A&E attendances. A similar proportion of CEC and CGP have received at least one prescription during the study period, but the average number of prescriptions is higher for CEC, meaning that, on average, CEC received more prescriptions.

The next tables give the most common prescriptions (table 5), outpatient clinic specialties (table 6) and diagnosed condition for acute inpatient admissions (table 7) (for coding schemas, see online supplemental table 5). The outcomes are ordered from largest to smallest by the total number of events for each health outcome and separately for the two cohorts. The resulting different ordering for the two cohorts is intentional and aims to highlight the very marked differences in health between the two cohorts. The tables clearly show that the reasons for contacting the health services are dissimilar between the two cohorts. For example, a higher proportion of the CEC cohort have had prescriptions for depression (CEC=19.7%, CGP=8.1%), psychiatric outpatient clinic attendances (CEC=20.1%, CGP=5.5%) and acute inpatient admissions due to mental and behavioural disorders (CEC=2.2%, CGP=0.3%). The proportion of CEC

**Table 5** PIS—Prescribing Information System

| BNF chapter/section | CGP Prescriptions N | % | Children N | % | Rate* | BNF chapter/section | CEC Prescriptions N | % | Children N | % | Rate* |
|---|---|---|---|---|---|---|---|---|---|---|---|
| 3—Respiratory system | 3 350 333 | 19.6 | 334 231 | 51.4 | 689.1 | 4—Central nervous system | 116 988 | 26.8 | 9215 | 66.6 | 1265.2 |
| 301—Bronchodilators | 1 164 625 | 6.8 | 130 986 | 20.2 | 241.5 | 407—Analgesics | 31 394 | 7.2 | 7570 | 54.7 | 393.4 |
| 304—Antihistamines | 1 108 966 | 6.5 | 236 979 | 36.5 | 215.6 | 404—CNS and ADHD | 28 455 | 6.5 | 794 | 5.7 | 224.3 |
| 302—Corticosteroids | 750 694 | 4.4 | 76 544 | 11.8 | 157.1 | 403—Antidepressant | 20 922 | 4.8 | 2723 | 19.7 | 260.2 |
| 13—Skin | 3 260 944 | 19.1 | 456 524 | 70.3 | 690.9 | 3—Respiratory system | 71 972 | 16.5 | 7537 | 54.5 | 713.4 |
|  |  |  |  |  |  | 301—Bronchodilators | 27 163 | 6.2 | 2897 | 20.9 | 265.5 |
| 4—Central nervous system | 2 301 960 | 13.5 | 327 896 | 50.5 | 530.5 | 302—Corticosteroids | 16 881 | 3.9 | 1625 | 11.7 | 169.9 |
| 407—Analgesics | 1 020 520 | 6.0 | 279 953 | 43.1 | 245.1 | 304—Antihistamines | 18 663 | 4.3 | 4779 | 34.6 | 169.5 |
| 403—Antidepressant | 398 369 | 2.33 | 52 768 | 8.1 | 114.4 |  |  |  |  |  |  |
| 404—CNS and ADHD | 273 164 | 1.6 | 7548 | 1.2 | 41.16 | 13—Skin | 62 960 | 14.4 | 9707 | 70.2 | 621.5 |
| 5—Infections | 2 234 038 | 13.1 | 470 671 | 72.4 | 487.8 | 5—Infections | 45 127 | 10.3 | 9520 | 68.8 | 448.8 |
| 501—Antibacterial drugs | 2 055 439 | 12.0 | 454 724 | 70.0 | 444.7 | 501—Antibacterial drugs | 41 721 | 9.5 | 9178 | 66.4 | 411.0 |
| 703—Contraceptives† | 779 348 | 4.6 | 133 343 | 41.7 | 349.0 | 1001—Drugs used in | 15 104 | 3.5 | 6041 | 43.7 | 146.3 |
| 1001—Drugs used in | 638 973 | 3.7 | 258 205 | 39.7 | 132.6 | rheumatic diseases gout |  |  |  |  |  |
| rheumatic diseases & gout |  |  |  |  |  | 601—Diabetes drugs | 14 391 | 3.3 | 139 | 1.0 | 124.9 |
| 601—Diabetes drugs | 592 741 | 3.5 | 5396 | 0.8 | 105.5 | 703—Contraceptives† | 11 552 | 2.6 | 2685 | 42.8 | 218.4 |

BNF sections are indented.

*Age standardised for age groups 0–24 using the 2013 European standard population per 1000 person-years.

†For contraceptives, the percentage of children and the age-standardised rate are calculated for the female members of the cohort.

ADHD, attention deficit hyperactivity disorder; BNF, British National Formulary; CNS, central nervous system.

**Table 6** SMR00—Outpatient Attendances

| Clinic specialty | CGP Attendances N | % | Children N | % | Rate* | Clinic specialty | CEC Attendances N | % | Children N | % | Rate* |
|---|---|---|---|---|---|---|---|---|---|---|---|
| D5—Orthodontics | 492407 | 20.4 | 51950 | 8.0 | 67.5 | Psychiatry (all) | 20170 | 26.5 | 2781 | 20.1 | 163.2 |
| C8—Trauma & Orthopaedic Surgery | 318094 | 13.2 | 121722 | 18.7 | 58.8 | C8—Trauma & Orthopaedic Surgery | 7507 | 9.8 | 2805 | 20.3 | 68.4 |
| AF—Paediatrics | 233131 | 9.7 | 66860 | 10.3 | 49.3 | AF—Paediatrics | 7015 | 9.2 | 2034 | 14.7 | 79.7 |
| A7—Dermatology | 203359 | 8.4 | 61886 | 9.5 | 40.3 | D5—Orthodontics | 6180 | 8.1 | 851 | 6.2 | 42.2 |
| Psychiatry (all) | 195907 | 8.1 | 35802 | 5.5 | 32.6 | Obstetrics† (all) | 5868 | 7.7 | 1113 | 17.7 | 131.6 |
| C5—Ear, Nose & Throat | 133917 | 5.6 | 63502 | 9.8 | 32.2 | C5—Ear, Nose & Throat | 3088 | 4.1 | 1452 | 10.5 | 38.0 |
| Dentistry (all) | 104934 | 4.4 | 24581 | 3.8 | 22.9 | Dentistry (all) | 2957 | 3.9 | 797 | 5.8 | 30.8 |
| C7—Ophthalmology | 80998 | 3.4 | 32000 | 4.9 | 21.7 | T2—Midwifery† | 2694 | 3.5 | 255 | 4.1 | 65.4 |
| A9—Gastroenterology | 47755 | 2.0 | 14737 | 2.3 | 10.8 | A7—Dermatology | 2602 | 3.4 | 858 | 6.2 | 23.5 |
| Obstetrics† (all) | 47015 | 2.0 | 12135 | 3.8 | 26.2 | G5—Learning Disability | 2134 | 2.8 | 195 | 1.4 | 29.9 |
| AH—Neurology | 46675 | 1.9 | 17394 | 2.7 | 9.4 | (Mental Handicap) | | | | | |
| F2—Gynaecology† | 45324 | 1.9 | 25508 | 8.0 | 23.6 | C7—Ophthalmology | 2067 | 2.7 | 869 | 6.3 | 36.4 |
| C9—Plastic Surgery | 44880 | 1.9 | 16511 | 2.5 | 8.8 | F2—Gynaecology† | 1844 | 2.4 | 908 | 14.5 | 44.1 |
| T2—Midwifery† | 20510 | 0.9 | 2561 | 0.4 | 11.3 | AH—Neurology | 1593 | 2.1 | 555 | 4.0 | 15.0 |
| G5—Learning Disability | 5033 | 0.2 | 1026 | 0.2 | 1.4 | C9—Plastic Surgery | 1070 | 1.4 | 393 | 2.8 | 11.0 |
| (Mental Handicap) | | | | | | A9—Gastroenterology | 725 | 1.0 | 273 | 2.0 | 6.8 |

*Age standardised for age groups 0–24 using the 2013 European standard population per 1000 person-years.
†For obstetrics, gynaecology and midwifery, the percentage of children and the age-standardised rate are calculated for the female members of the cohort.
CEC, care experienced children; CGP, children in the general population.

**Table 7** SMR01 —General/Acute Inpatient and Day Cases by ICD-10 chapters

CGP

| Main diagnosis | Attendances | | Children | | |
|---|---|---|---|---|---|
| | N | % | N | % | Rate* |
| Diseases of the digestive system | 61704 | 15.5 | 41483 | 6.4 | 17.0 |
| Dental caries (K02) | 14084 | 3.5 | 13301 | 2.0 | 7.0 |
| Injury, poisoning and other consequences of external causes | 55038 | 13.8 | 42002 | 6.5 | 12.0 |
| Other injury | 28726 | 7.2 | 22899 | 3.5 | 6.3 |
| Head injury | 11095 | 2.8 | 9745 | 1.5 | 2.7 |
| Drug poisoning | 9251 | 2.3 | 6129 | 0.9 | 1.8 |
| Hand injury | 5966 | 1.5 | 5435 | 0.8 | 1.3 |
| Symptoms, signs and abnormal clinical and laboratory findings | 50192 | 12.6 | 33271 | 5.1 | 12.0 |
| Diseases of the respiratory system | 39226 | 9.8 | 25628 | 3.9 | 10.5 |
| Asthma | 9172 | 2.3 | 7562 | 1.2 | 2.7 |
| Diseases of the genitourinary system | 25776 | 6.5 | 16600 | 2.6 | 5.4 |
| Diseases of the musculoskeletal system and connective tissue | 23431 | 5.9 | 12365 | 1.9 | 4.5 |
| Neoplasms | 21800 | 5.5 | 5340 | 0.8 | 4.9 |
| Examination, observation, etc | 19259 | 4.8 | 12918 | 2.0 | 4.4 |
| Obstetric† | 15099 | 3.8 | 13089 | 4.1 | 6.9 |
| Medical abortion (O04)† | 13574 | 3.4 | 12107 | 3.8 | 6.1 |
| Endocrine, nutritional and metabolic diseases | 13910 | 3.5 | 3978 | 0.6 | 2.6 |
| Mental and behavioural disorders | 2646 | 0.664 | 2207 | 0.3 | 0.5 |

CEC

| Main diagnosis | Attendances | | Children | | |
|---|---|---|---|---|---|
| | N | % | N | % | Rate* |
| Injury, poisoning and other consequences of external causes | 3900 | 27.7 | 2198 | 15.9 | 36.2 |
| Drug poisoning | 1671 | 11.9 | 849 | 6.1 | 13.0 |
| Other injury | 1262 | 9.0 | 895 | 6.5 | 10.7 |
| Head injury | 626 | 4.5 | 514 | 3.7 | 5.2 |
| Hand Injury | 341 | 2.4 | 298 | 2.2 | 2.9 |
| Symptoms, signs and abnormal clinical and laboratory findings | 1811 | 12.9 | 1059 | 7.7 | 17.3 |
| Diseases of the digestive system | 1626 | 11.6 | 1158 | 8.4 | 27.2 |
| Dental caries (K02) | 631 | 4.5 | 590 | 4.3 | 17.7 |
| Diseases of the respiratory system | 1108 | 7.9 | 618 | 4.5 | 12.4 |
| Asthma | 317 | 2.3 | 116 | 0.8 | 2.9 |
| Diseases of the genitourinary system | 867 | 6.2 | 463 | 3.3 | 7.3 |
| Obstetric† | 702 | 5.0 | 539 | 8.6 | 15.5 |
| Medical abortion (O04)† | 538 | 3.8 | 447 | 7.1 | 11.3 |
| Endocrine, nutritional and metabolic diseases | 604 | 4.3 | 114 | 0.8 | 5.7 |
| Examination, observation, etc | 566 | 4.0 | 426 | 3.1 | 5.1 |
| Diseases of the musculoskeletal system and connective tissue | 401 | 2.9 | 234 | 1.7 | 5.9 |
| Mental and behavioural disorders | 399 | 2.8 | 310 | 2.2 | 3.0 |
| Neoplasms | 168 | 1.195 | 80 | 0.6 | 1.4 |

Subgroups within ICD-10 chapters are indented.

*Age standardised for age groups 0–24 using the 2013 European standard population per 1000 person-years.

†For obstetrics and medical abortion, the percentage of children and the age-standardised rate are calculated for the female members of the cohort.

CEC, care experienced children; CGP, children in the general population.

hospitalised due to injuries, drug poisoning and other external causes is also higher compared with CGP (for drug poisoning CEC=6.1%, CGP=0.9%). The differences between the two cohorts in age-standardised rates for these prescriptions and hospitalisations are similarly notable.

A higher proportion of care experienced young women have had outpatient obstetrics (CEC=17.7%, CGP=3.8%), gynaecology (CEC=14.5%, CGP=8.0%) and midwifery (CEC=4.1%, CGP=0.4%) clinic attendances, and have had an abortion (CEC=7.1%, CGP=3.8%). The age-standardised rates for these hospital attendances are also higher for the care experienced young women compared with the women in the general population. The proportion of care experienced young women who have been prescribed contraceptives is similar to that of young women in the general population (CEC=42.8%, CGP=41.7%); however, the total number of contraceptive prescriptions and the age-standardised rates are higher for the women in the general population ($CEC_{rate}$=218.4, $CGP_{rate}$=349.0). Without additional data on the type of contraceptive prescribed (long-acting, eg, implants, vs not long-acting, eg, oral contraceptives), we cannot say what drives this difference. However, previous research suggests that women from more deprived areas are more likely to receive long-acting reversible contraceptives,[25] and this could explain why the care experienced women in our cohort have fewer prescriptions compared with the women in the general population.

These initial results show substantial differences in health and health service use between the two cohorts. First, the proportion of children who have been in contact with the health services is higher for the CEC compared with the CGP. Second, CEC have more frequent contact with health services across all data sets. Third, many of the reasons for contacting health services are different for the two cohorts, with the CEC having more frequent contact related to mental, sexual and reproductive health. The level of contact with the health services for some causes (eg, outpatient attendance for ear, nose and throat diseases) is similar for the two cohorts.

The CHiCS cohorts outlined here show substantial differences in the socioeconomic background and health between children who have experienced care and those who have not. They provide a unique opportunity for the first population-wide evidence on objective health outcomes for an understudied and vulnerable population of children. Next, our research will focus on explaining the differences in the health outcomes between the two cohorts of children using event history analysis on matched cohorts.

### Strengths and limitations

The CHiCS project is the first national longitudinal data collection in the UK that compares the health of CEC with CGP. In addition to the large population-wide sample and a 7-year follow-up, other main strengths of the study include the wide range of health outcomes available,

along with the high quality and representativeness of the data. Together, these will allow for robust and detailed results on a very vulnerable population group.

The weaknesses of the study include errors in the individual data sets and in data linkage. Of the health data sets, the SMR00 (outpatient attendances) is of weakest quality with more inconsistent or inaccurate recording of sex. However, the noted errors in recording data and in linkage only affect approximately 3% of children in both cohorts and for the rest of the children we did not note any inconsistencies (eg, in sex or age) across the data sets.

Another weakness is that the PC excludes children educated at home or at independent schools. However, only about 25–30 000 children are educated in independent schools (4% of all children attending school in Scotland).[26] Based on this and on the comparison of our data to the population census estimates of the number of children in Scotland (online supplemental table 5), we are confident that our data are representative of school children in Scotland.

Additional potential confounders on parents' background over and above what has been presented (ie, mother's age at birth, parental employment status at birth, area of residence at birth) were not available for this project. However, future research may be able to link maternal health to these data through Birth Registrations.

As this study has highlighted mental, sexual and reproductive health as areas with biggest health differences between the two cohorts, future research would benefit from more focused attention and detailed data collection in a specific area of health. This could include more detailed data on prescriptions (eg, to distinguish long-acting and other contraceptives) and, where possible, linking in smaller less often used community health or primary care data sets.

### Collaboration

To access the data, the authors applied for data access to the Public Benefit and Privacy Panel for Health and Social Care and to the Scottish Government's Statistics Public Benefit and Privacy Panel. After data access was approved by both organisations, a data sharing agreement between the University of Glasgow and the Scottish Government was agreed. Due to the sensitivity of the data, it will not be made publicly available.

We encourage collaboration from other researchers, charities and public bodies interested in this research that fit with the original aims of this study. We also encourage collaborations to extend the follow-up period of both cohorts to 2020 and beyond to study the effects of national lock-downs on the health of young adults and care leavers.

**Acknowledgements** The authors would like to acknowledge the support of the eDRIS Team (Public Health Scotland) for their involvement in obtaining approvals, provisioning and linking data and the use of the secure analytical platform

within the National Safe Haven. The National Records of Scotland undertook the indexing and provided the linkage key for this study. The authors would also like to acknowledge the Scottish Exchange of Data (ScotXed) for allowing us to access their data and the CHiCS project Advisory Group members for their engagement and feedback on the initial study results.

**Contributors** MA and DB conceived the research idea and carried out most of the research and writing. CTBL and CM assisted in initial data analysis and provided feedback on the draft of this paper. AHL and MH contributed to the research design and have provided important feedback throughout the project and writing of this paper.

**Funding** This work was supported by the Economic and Social Research Council (grant number ES/T000120/1). The data linkage and access to the National Safe Haven was facilitated by the Urban Big Data Centre (grant number ES/L011921/1). MA, DB and AHL are also funded by the Medical Research Council (MC_UU_00022/2) and the Scottish Government Chief Scientist Office (SPHSU17).

**Competing interests** None declared.

**Patient consent for publication** Not required.

**Ethics approval** Ethical approval was obtained from the University of Glasgow College of Medicine, Veterinary and Life Sciences Ethics Committee (Project No: 200160031).

**Provenance and peer review** Not commissioned; externally peer reviewed.

**Data availability statement** Data may be obtained from a third party and are not publicly available. These data can be accessed through applications to the Public Benefit and Privacy Panel for Health and Social Care (https://www.informationg overnance.scot.nhs.uk/pbpphsc/https://www.informationgovernance.scot.nhs.uk/pbpphsc/) and to the Scottish Government's Statistics Public Benefit and Privacy Panel (https://www.gov.scot/publications/scottish-government-statistics-request-our-data/).

**ORCID iD**
Mirjam Allik http://orcid.org/0000-0003-1674-3469

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
