## [Reviewer comments · BMJ Open]

ARTICLE DETAILS

TITLE (PROVISIONAL)	Cohort profile: The “Children’s Health in Care in Scotland” (CHiCS) study – a longitudinal dataset to compare health outcomes for care experienced children and general population children.
AUTHORS	Allik, Mirjam; Brown, Denise; Taylor Browne Luka, Courtney; MacIntyre, Cecilia; Leyland, Alastair; Henderson, Marion

VERSION 1 – REVIEW

REVIEWER	Liu, Can Centre for Health Equity Studies
REVIEW RETURNED	06-Jul-2021

GENERAL COMMENTS	Thank you for inviting me to review this cohort profile. I think it is an important data source that will contribute to the understanding of adverse social and health outcomes of children experienced care. I have some general questions along with specific suggestions. Title: 1. The title is very long and hard to follow for readers who are not familiar with the names of the data sources. I suggest trim down and to make the cohort (care experienced children and general population) stand out. Abstract: 2. Why CEC are defined only from those were on CLAS 2009/2010 while CGP are defined as those not on CLAS 2007-2016. Why not look into CEC in 2007-2008 and 2011-2016?3. Should the bullet points be separated for “Article summary” and “Strengths and limitations of the study”? if so, perhaps the second point belong to the strengths? Introduction: 4. Page 4 line 26-28. This first sentence of the paragraph is pointing on the limitation of existing data and I was expecting some solution following it. Perhaps it can be reorganized to align with some of the following paragraph. Cohort description: 5. I did not find description on the data source for the health outcomes and the quality of the data.6. Please specify the age range of school-aged children, as the school-age may differ by country.7. I wonder if there are linkage between non-twin siblings. The family clustering may need to be taken into consideration for future studies.8. Health Board is not familiar. It seems to mean geographical units. Does the health board in charge of organizing the health or social care?
---

	9. I wonder if the quality of the pupil census is dependent on characteristics of the school or region. As it seems to be the basis for defining the population, it should include all pupils irrespective of their CEC status, private/public school, or other socioeconomic characteristics that may differ between CES and CGP. If there is any selection into observation, it should be specified. 10. What does residential stability add? Isn't it natural that if the child was in care, the residential zone would change (eg. Move to foster care family)? 11. Page 7 line 19 number before % was missing Table 5-6. 12. it seems the clinic specialty are ordered by prevalence in CGP and CEC respectively. Surely this is informative but also made it harder to compare across the two cohorts. Suggest making the specialty align in both groups and add another column indicating the rank of commonality. Figure 1. 13. What are the indices? It was never mentioned in the main text. General comments: 14. Abbreviations are too many, to the extent that making it difficult for readers to follow. Perhaps CEC and CGP can be spelled out throughout as the abbreviations did not save too much words. Also make sure that once the abbreviation is specified, it is consistently used, e.g. care experienced children were still used after CEC in abstract and introduction. 15. I would appreciate some description on the age range of observed outcomes. It is clear when, in the calendar year, the data was collected. But I wonder how old were the children at school-age (as commented above) and at what age were the follow up on the outcomes stopped.
--	--

REVIEWER	Vinnerljung, Bo Stockholm University, Department of Social Work
REVIEW RETURNED	15-Jul-2021

GENERAL COMMENTS	The manuscript is basically a report on the first summary results of a longitudinal large sample project investigating health outcomes for children with out-of-home care experiences (OHC). Studies employing national cohort design for examining outcomes of OHC has mostly been a reserve for Nordic scholars. Therefore this project from Scotland is a welcome and praiseworthy addition to the research field. The design and the initial results reported in the manuscript shows great promise, in my opinion. However, I have a few concerns that - in the light of the Nordic studies in this field - should be better described in this manuscript, which is a bit of a first installment in what will probably be a long row of studies. 1. What data are available on placement history, and how are the authors planning to use these data in the analyses? Time spent in care during the formative years is a must variable in order to explore the heterogeneity of the OHC population. Without good data on time in care, and preferably also stability, the OHC experience will mainly be a marker of a vulnerable childhood. In order to estimate the impact of OHC on health, especially long term impact, analyses will - as a first step - have to look at eg. those children who have been in long-term care.
--

	2. Confounders: the SES-data seem excellent (including area data). But the Nordic studies have also shown the strong influence of individual parental data, eg. mental ill health, substance abuse and crime. What confounders are the authors planning to use in the forthcoming analyses? Which are available, and which are not?
--	---

VERSION 1 – AUTHOR RESPONSE

Thank you for the feedback on our paper, it has been very helpful in improving our work. We have responded to your comments below, with your comments in italics and our response in normal font. The changes in the text are highlighted with bold font. We hope you find the revised paper an improvement.

Reviewer 1:

1. The title is very long and hard to follow for readers who are not familiar with the names of the data sources. I suggest trim down and to make the cohort (care experienced children and general population) stand out.

The title is now shortened and revised.

2. Why CEC are defined only from those were on CLAS 2009/2010 while CGP are defined as those not on CLAS 2007-2016. Why not look into CEC in 2007-2008 and 2011-2016?

Both of our cohorts, the care experienced and the general population, are children who were in school in 2009 and both were then followed over a 7-year period. The comparison to the 2007-2016 social care records for the general population is only made to ensure that none of those children were or became looked after. This was done to ensure that the general population cohort does not include anyone with a care history. Looking at our wording in the paper on page 5, it could have been presented better. We have adjusted the wording and hope it is clearer now why the comparison was made to all records between 2007-2016. We have left the wording in the abstract as is.

3. Should the bullet points be separated for “Article summary” and “Strengths and limitations of the study”? if so, perhaps the second point belong to the strengths?

We have made changes as per editorial recommendations (see below). There is a new “strengths and limitations” section added on page 9.

4. Page 4 line 26-28. This first sentence of the paragraph is pointing on the limitation of existing data and I was expecting some solution following it. Perhaps it can be reorganized to align with some of the following paragraph.

We have removed the first sentence and placed it in the next paragraph with minor rewording.

5. I did not find description on the data source for the health outcomes and the quality of the data.

The health data was briefly described at the start of the results section, but this has now been moved to the cohort description where the data linkage is described. A comment and references on data quality have been added.

6. Please specify the age range of school-aged children, as the school-age may differ by country.

Children in Scotland usually start primary school at age four and a half to five and a half years old. Secondary school begins at age 11 or 12 for a compulsory four years with the following two years being optional. We have now clarified this in the paper (see Cohort description, 1st paragraph).

7. I wonder if there are linkage between non-twin siblings. The family clustering may need to be taken into consideration for future studies.

Linkages between family members/siblings are theoretically feasible through birth records and we understand that there is very preliminary research into the feasibility and quality of this. However, this is still at very early stages.

8. Health Board is not familiar. It seems to mean geographical units. Does the health board in charge of organizing the health or social care?

Health Boards (14 in total) are responsible for delivering health care in Scotland and are made up of local authorities. Social care is provided by local authorities (32 in Scotland). While conducting research we looked at differences by local authorities also, but only included results by Health Boards in the Supplement because some local authorities are very small and may pose a disclosure risk. We have now described Health Boards in more detail in the paper (see Area level socioeconomic profile, 2nd paragraph).

9. I wonder if the quality of the pupil census is dependent on characteristics of the school or region. As it seems to be the basis for defining the population, it should include all pupils irrespective of their CEC status, private/public school, or other socioeconomic characteristics that may differ between CES and CGP. If there is any selection into observation, it should be specified.

The Pupil Census has nearly 100% coverage of children attending publicly funded schools and only excludes private schools (about 4% of the total student population – see added strengths and weaknesses section on page 9). It is used in accredited national statistics and frequently also in research. There is no selection into the census and the quality is reported very high across schools/regions. We have included a statement to that effect on page 5. There is no publicly available quality statement to reference, but its use in accredited national statistics should speak to the quality.

10. What does residential stability add? Isn't it natural that if the child was in care, the residential zone would change (eg. Move to foster care family)?

Yes, we would have expected children in care to be more residentially mobile overall. However, this is not necessarily the case for all children in care in Scotland as some children remain looked after at home (under a supervision requirement, with social care visits made to home). We decided to include this Table in the Supplement to give a sense of the residential mobility rate in the group of children with care experience but also to highlight residential mobility rates of children in Scotland more generally. This Table also works well to highlight the problem of missing data in this group of children. In total 10% of care experienced children did not have residential details recorded in the 2009 Pupil Census, compared to 0.04% in the general population. We have made this point more clearly in the paper (see Area level socioeconomic profile, 2nd paragraph).

11. Page 7 line 19 number before % was missing.

Sentence wording fixed.

12. Tables 5-6. It seems the clinic specialty are ordered by prevalence in CGP and CEC respectively. Surely this is informative but also made it harder to compare across the two cohorts. Suggest making the specialty align in both groups and add another column indicating the rank of commonality.

We understand the comment but have decided to keep the tables as is. With this presentation style we hope to further underline the differences in health outcomes between the two cohorts. We partly want to make this comparison uncomfortable as the results themselves are uncomfortable.

13. Figure 1. What are the indices? It was never mentioned in the main text.

We have included an explanation in the footnote of the figure.

14. Abbreviations are too many, to the extent that making it difficult for readers to follow. Perhaps CEC and CGP can be spelled out throughout as the abbreviations did not save too much words. Also make sure that once the abbreviation is specified, it is consistently used, e.g. care experienced children were still used after CEC in abstract and introduction.

We recognize that we have used many abbreviations, partly this is due to the paper being very technical and describing a linkage of 10 data sources, the multiple linkage steps, variables and institutions involved in this. We have used CEC throughout the paper now and only use care “experienced population/young women/youth” where it is necessary to refer to a care experienced group that could not be described as “children”.

15. I would appreciate some description on the age range of observed outcomes. It is clear when, in the calendar year, the data was collected. But I wonder how old were the children at school-age (as commented above) and at what age were the follow up on the outcomes stopped.

Children were aged four to 17 years old at the start of the study and 11 to 24 years old by the end of follow-up in 2016. We have made this clearer in the paper (see Cohort description, last paragraph, page 6).

Reviewer 2:

1. What data are available on placement history, and how are the authors planning to use these data in the analyses? Time spent in care during the formative years is a must variable in order to explore the heterogeneity of the OHC population. Without good data on time in care, and preferably also stability, the OHC experience will mainly be a marker of a vulnerable childhood. In order to estimate the impact of OHC on health, especially long term impact, analyses will - as a first step - have to look at eg. those children who have been in long-term care.

This was an oversight on our part. We have added a paragraph on the data we have available on care histories on page 6 and added a table to show the distribution of care placements and their lengths by type.

2. Confounders: the SES-data seem excellent (including area data). But the Nordic studies have also shown the strong influence of individual parental data, eg. mental ill health, substance

abuse and crime. What confounders are the authors planning to use in the forthcoming analyses? Which are available, and which are not?

Unfortunately, we do not have any additional parental information over and above what has been presented in the cohort profile paper (i.e. mother's age at birth, parental employment status at birth, area of residence at birth). Future research may be able look into this by linking in maternal health through birth registrations. It is questionable whether paternal information can be linked. In terms of data on children themselves, we have geographical information about area of residence and school attended. We also have information on prior health (through previous hospital records) which include information on e.g. psychiatric admissions, but also on long-term health conditions (epilepsy, diabetes), so will be able to adjust for this in future analyses. For care experienced children we also have detailed information on care histories.

VERSION 2 – REVIEW

REVIEWER	Liu, Can Centre for Health Equity Studies
REVIEW RETURNED	31-Aug-2021
GENERAL COMMENTS	Thanks. I have no further comments.
REVIEWER	Vinnerljung, Bo Stockholm University, Department of Social Work
REVIEW RETURNED	17-Aug-2021
GENERAL COMMENTS	In my view, the comments from the reviewers have been reasonably addressed. The added section on strengths and limitations provided increased clarity. I have no further comments.